# Effects of Florivory on Floral Visitors and Reproductive Success of *Sagittaria lancifolia* (Alismataceae) in a Mexican Wetland

**DOI:** 10.3390/plants13040547

**Published:** 2024-02-17

**Authors:** Dulce Rodríguez-Morales, Armando Aguirre-Jaimes, José G. García-Franco

**Affiliations:** 1Instituto de Neuroetología, Universidad Veracruzana, Xalapa 91190, VC, Mexico; dulcrodriguez@uv.mx; 2Red de Interacciones Multitróficas, Instituto de Ecología, A.C., Carretera Antigua a Coatepec 351, El Haya, Xalapa 91070, VC, Mexico; armando.aguirre@inecol.mx; 3Red de Ecología Funcional, Instituto de Ecología, A.C., Carretera Antigua a Coatepec 351, El Haya, Xalapa 91070, VC, Mexico

**Keywords:** flower damage, plant–animal interactions, floral visitors, plant fitness

## Abstract

Florivores consume floral structures with negative effects on plant fitness and pollinator attraction. Several studies have evaluated these consequences in hermaphroditic plants, but little is known about the effects on monoecious and dioecious species. We characterize the florivory and its effects on floral visitors and reproductive success in a monoecious population of *Sagittaria lancifolia*. Five categories of florivory were established according to the petal area consumed. Visits were recorded in male and female flowers within the different damage categories. Reproductive success was evaluated through fruit number and weight, as well as the number of seeds per fruit. Our results show that the weevil *Tanysphyrus lemnae* is the main florivore, and it mainly damages the female flowers. Hymenoptera were recorded as the most frequent visitors of both male and female flowers. Male and female flowers showed differences in visit frequency, which decreases as flower damage increases. Reproductive success was negatively related to the level of damage. We found that florivory is common in the population of *S. lancifolia*, which can exert a strong selective pressure by making the flowers less attractive and reducing the number of seeds per fruit. Future studies are needed to know how florivores affect plant male fitness.

## 1. Introduction

Florivory is a type of antagonistic interaction in which herbivores consume the reproductive structures of the plant [1,2,3,4,5,6]. Floral structures have complex and complementary functions in plant reproduction, so any damage caused by florivores can have serious consequences on plant fitness [6,7]. It has been stated that florivory has both direct and indirect negative effects on plant fitness: (a) directly, by consumption of the stamens and style, limiting the export of the plant’s own pollen and reception of external pollen, and by consuming the ovules that reduces the number of seeds produced [8,9,10,11]; and (b) indirectly, by altering the quality and quantity of the essential floral attributes that drive pollination, since by modifying the size and morphology of the flowers, the attraction and recognition of pollinators are reduced [2,3,9,12,13,14,15]. Several studies have shown that florivory reduces pollinator visits and therefore the production of fruits and seeds, since the pollinators prefer to visit intact flowers [4,6,9,16,17,18,19,20,21,22,23,24,25].

The magnitude of the effect of florivory could vary depending on the particular flower structures consumed but it can also differ according to the sexual expression and mating systems of the flowers themselves [21]. Sexual systems are very diverse in plants, although the most common is hermaphroditism [26], and seed production ranges from primarily self-fertilization to exclusively outcrossing [27]. Hermaphroditic and self-compatible plants might not be affected to the same extent by consumption of the petals, since florivory could act to increase self-pollination [28], but in plants with separate sexes such as the dioecious or monoecious species that require floral visitors to achieve fertilization, florivory can reduce reproductive success [6,25,29,30,31,32,33]. However, the effects of different levels of florivory have been minimally studied in the field [6], and little is known about the effect of this process on plants with other sexual expressions [34,35,36].

We studied a monoecious population of *Sagittaria lancifolia* in a wetland on the coast of the Gulf of Mexico. Specifically, the following aspects were assessed under natural conditions: (a) we identified the main florivore and characterized the damage in both female and male flowers; (b) we identified and quantified the frequency of floral visitors in both flower types with different degrees of damage; and finally, (c) we determined the impact of the degree of flower damage on the frequency of floral visitors and reproductive success of *S. lancifolia*.

## 2. Results

### 2.1. Florivores and Floral Damage

In La Mancha in Veracruz, Mexico, the flowers of *Sagittaria lancifolia* are consumed mainly by the weevil *Tanysphyrus lemnae* (Curculionidae) and, to a lesser extent, by grasshoppers (Orthoptera, Acrididae) (Figure 1A). Florivory was characterized in 585 flowers of *S. lancifolia* (276 females and 309 males). We recorded 206 flowers within the category of 0% damage, 371 flowers had between 1 to 50% damage, and 8 flowers presented more than 50% of the petal area removed. We recorded more male flowers without damage (121 flowers) than females (85 flowers). The same pattern was repeated in the categories with less than 12% floral damage, while in the categories between 13% and 100% of the petal area removed, the female flowers presented more damage than the male flowers. In the category with more than 50% of the petal area removed, we found only female flowers (8 flowers) (Figure 2). The Florivory Index was higher in the female (1.97) than in the male (1.49) flowers, and we found that the percentage of petal area removed was greater in female (5.58 ± 0.89; mean ± SE) than in male (3.92 ± 0.67) flowers (Wilcoxon test = 1650; *p* = 0.005).

### 2.2. Florivory and Floral Visitors

The flowers are visited by diverse orders of insects (Figure 1B). The total number of floral visits to *Sagittaria lancifolia* recorded was 1808, of which 702 were to female flowers, and 1106 to male flowers. The floral visitors belong to the orders Coleoptera, Diptera, Hemiptera, Hymenoptera, and Lepidoptera. Hymenoptera was the order with the highest frequency of visits to female (36.7%) and male (63.7%) flowers, followed by Diptera (12.6% and 2.9%, respectively) and Lepidoptera (2.2 and 1.6%, respectively). Coleoptera visited more female (3.1%) than male (0.2%) flowers. Some differences were observed in the total number of visits between flowers of different sexes, as well as among damage categories (Figure 3). There was a significant effect of flower sex (*F* = 13.2, *p* = 0.001), florivory damage (*F* = 9.08, *p* = 0.001), and their interaction on the frequency of flower visitors (*F* = 3.742, *p* = 0.002). In general, the highest values of visits were recorded in male flowers. Undamaged male flowers presented more visits than their female counterparts. Both sexes of flowers presented the same pattern, with no significant differences in the number of visits from the no-damage up to the 25% damage category. The number of floral visits was significantly lower when male and female flowers presented the highest percentage of damage.

### 2.3. Florivory and Female Reproductive Success

In general, the largest number of fruits developed was found in the low damage categories <12% (Figure 4A). The fruit set recorded ranged between 61 and 89%. However, a high number of developed fruits was recorded in the undamaged flowers (84%), and in those comprising the three lowest florivory damage categories (fruit set 81–89%). The lowest fruit set was found in the highest categories of florivory damage (Figure 4B).

Fruit weight was negatively affected by florivory. The fruits from undamaged flowers and those in the categories with the lowest florivory damage (0 to 25% of damage) were heavier than the fruits developed from flowers in the two categories of the highest florivory damage (H = 32.4, *p* < 0.001) (Table 1). The weight of the seeds per fruit presented a similar pattern. The seeds of fruits developed in flowers with little or no damage were heavier than those of fruits developed in flowers with the greatest amount of damage (H = 30.5, *p* < 0.001). The numbers of seeds developed in the fruits of the different damage categories were different (H = 14.8, *p* = < 0.012). The highest number of seeds was recorded in the flowers without damage (1287.3 ± 107.7, mean ± SE), and the average number of seeds per fruit decreased as the percentage of florivory increased (Table 1). Our results with manual damage of the flowers showed that the weight (t-student = 0.075, *p* = 0.94) and number (t-student = 0.398, *p* = 0.69) of seeds obtained from fruits developed in manually pollinated flowers without damage and with a high level of florivory did not differ (Table 2).

## 3. Discussion

In this study, we found that florivory in *S. lancifolia* is caused by insects, mainly the weevil *Tanysphyrus lemnae*. Beetles are the most diverse group of organisms on the planet, with around 300,000 to 450,000 species, covering a very important spectrum of functional groups [37]. The main families reported as florivores include Chrysomelidae (32,500 species) and Curculionidae (51,000 species) [37]. The presence of these beetles as potential florivores is because some species use part of the floral structures as a food resource or develop part of their life cycle within the flower buds [38]. For example, *Lepidium papilliferum* (Brassicaceae) presents damage to the petals caused by *Phyllotreta* sp. (Chrysomelidae) [16]. In populations of *Solanum rostratum* in Mexico, several species of Chrysomelidae and Scarabaeidae are reported to consume parts of the corolla, anthers, and stamens [23]. Likewise, *Sennius* sp. (Chrysomelidae) has been reported in *Chamaecrista cathartica* (Fabaceae), developing part of its life cycle in the flower buds [15], larvae of *Anthonomus signatus* (Curculionidae) develop in the flower buds and consume the pollen of *Fragaria virginiana* (Rosaceae) [28], and the larvae of *Cionus nigritarsis* (Curculionidae) develop inside the flower buds of *Verbascum nigrum* (Scrophulariaceae), consuming the floral and reproductive tissues [19]. Unlike those studies, in *S. lancifolia*, the weevil arrives at the flowers as an adult and consumes the petals and, to a lesser extent, the reproductive structures, producing the greatest impact on the corolla, which could negatively affect the subsequent attraction of pollinators.

In general, a wide variation has been documented in the incidence of florivory in plants from tropical and temperate zones (0.2 to 97%) and this can be double in tropical compared to temperate regions [6]. Likewise, solitary flowers present higher frequencies of attack than inflorescences, due in part to the fact that solitary flowers are larger than those grouped in inflorescences [6,39]. In our study, we found an incidence of florivory of almost 50%, i.e., half of the flowers in the inflorescence of *S. lancifolia* presented some degree of damage. Moreover, there is a greater intensity of florivory in female than in male flowers (5.58 and 3.92, respectively). In other species from aquatic environments, Ortiz et al. [39] report high incidence values of florivory for *Eichornia azurea* (93.3%), *Ludwigia lagunae* (96.7%), *L. tomentosa* (66.7%) and *Nimphae amazonium* (100%), as well as in *Hydrocleys parviflora* (55.5%), with values close to those found in our study. Likewise, variation in the intensity of herbivory has been found in aquatic species such as *Echinodorus grandiflorus* (1.7%), *Hydrocleys flava* (9.7%), *Pontederia rotundifolia* (15.4%), and *P. parviflora* (35.5%), among others [39]. Although there is variation in the incidence and intensity of florivory between environments and between terrestrial and aquatic species, no studies have evaluated these variables considering plant species with female and male flowers (dioecious or monoecious). Most studies have focused on species that have hermaphroditic flowers [2], and a few studies on gynodioecious species or plants with flowers of separate sexes (dioecious or monoecious) [6,9,40]. This suggests that future studies should focus on plant species with other sexual expressions, allowing us to identify patterns and differences among them.

Several studies have shown that the damage caused by herbivores affects the frequency of floral visitors through modification of the flower morphology [6,9,18,25], and it has been reported that the level of damage to the flowers plays an important role in flower recognition by pollinators [6]. Our data show that florivory intensity has a differential effect on visitation frequency in female and male flowers of *S. lancifolia*. As the damage to the flowers increases and their morphology changes, the number of floral visitors decreases. This negative effect is greater in the female flowers, particularly at higher levels of florivory. Some groups of insects visited flowers all along the gradient of damage levels, such as Hymenoptera, which was the order that generally had the highest number of visits, followed by Diptera, although visits by the latter were substantially lower. These two orders of insects have been recorded as the main floral visitors in other species of *Sagittaria* [12,41,42], which may be due to the generalist morphology of the flowers, with extended white petals, and because they offer rewards of nectar (both sexes) and pollen (male flowers only). Lepidoptera, which was generally rare in *S. lancifolia* flowers, did not visit the flowers in the two categories of the highest level of damage (26–100% damage). This suggests that individuals of Diptera and Lepidoptera are occasional visitors that could be acting as nectar robbers in the *S. lancifolia* flowers but that they are more susceptible than Hymenoptera to the flower’s morphological changes as a product of herbivory since these seem to reduce their ability to recognize the flowers. Specific studies on variation in the attraction of each group of floral visitors will allow us to more clearly understand the effect of the level of florivory in *S. lancifolia*.

Some studies have indicated the negative consequences of florivory for the reproductive success of several plant species [2,28,30,43,44], and most studies evaluated reproductive success at the level of fruit set [8,18,21,45,46]. Our study shows that the fruit set, fruit weight, and number of seeds per fruit are all reduced with increased levels of florivory of *S. lancifolia*. When the damage to the female flowers of *S. lancifolia* is low (≤12%), there is no negative effect on fruit production, weight, and number of seeds; but when florivory increases (≥13%), these variables of reproductive success are negatively affected, with fruit weight and seed number both decreasing by 49% in the highest category of damage, relative to the maximum fruit weight and seed number recorded. These findings indicate that the increase in florivory reduces the frequency of floral visitors, but also decreases the availability and load of pollen received by the damaged flowers, with a consequent reduction in fruit weight and number of seeds produced [3,45,47]. Florivores eat the petals, but also remove a substantial amount of pollen, reducing the pollen load that will ultimately be deposited in female flowers. Specific future studies are required to evaluate the extent to which the florivores reduce the pollen available in male flowers.

The variation in the incidence of florivory in the *S. lancifolia* flowers forced us to complete the set of experimental flowers of each category by producing artificial damage in some flowers. It has been reported that experimental manipulation of flowers could have a direct effect on fruit and seed production [3,48] since damage changes the reproductive response of flowers. Our manual pollination experiment (with pollen saturation) in undamaged and manually damaged female flowers at the maximum level of damage confirmed that the manipulation conducted on the flowers had no effect, since seed production and fruit weight were similar in the two treatments (intact flowers and with manual damage). This reinforces the notion that the reduction recorded in the frequency of floral visitors and reproductive success in *S. lancifolia* is a consequence of the different levels of florivory and supports the recorded negative effect of florivory on reproductive success in *Eriotheca gracilipes* and *Nemophila menziesii* [3,17].

## 4. Materials and Methods

### 4.1. Study Site

The study was conducted at the Centro de Investigaciones Costeras “La Mancha” (CICOLMA), located in the Center of the Gulf of Mexico coast, in the State of Veracruz, Mexico (19°40′33″ N–96°24′48″ W). The climate is warm and sub-humid with rain in the summer. The average annual rainfall is 1300 mm, and the mean annual temperature is 25 °C [49]. La Mancha comprises only 83.2 ha; however, it contains different plant communities: tropical sub-deciduous forest, tropical dry forest, mangrove forest, coastal dunes, scrub, beach vegetation, and wetland [50,51]. The La Mancha wetland, recognized by the RAMSAR Convention (La Mancha–El Llano No. 1336), is a site with high productivity and biodiversity, with species such as *Thypha domingensis* (Typhaceae), *Ponthederia sagittata* (Pontederiaceae) and *Sagittaria lancifolia* (Alismataceae) in flooded areas, and *Cyperus articulatus*, *Oxycaryum cubense*, *Eleocharis geniculata* (Cyperaceae) and *Annona glabra* (Annonaceae) in less flooded areas [52] (Figure 5A).

### 4.2. Study Species

*Sagittaria lancifolia* L. (Alistamataceae) is a perennial herb with clonal growth. Its leaves emerging from the water level are variable in size and shape, but the blades are mostly lanceolate and are 15–35 cm long. In Mexico, the species is distributed throughout the coastal plain of the Gulf of Mexico [41] and monoecious and dioecious plants have been reported [53], although the population at La Mancha is monoecious only. Flowering occurs throughout the year, but it peaks between June and August [52,54]. Each *S. lancifolia* plant has only one functional inflorescence at a time. The scape is erect up to 2 m long and the inflorescence is 16–55 cm long with 5–13 whorls of flowers, often branched at the bottom (Figure 1B). In La Mancha, the petals of staminate flowers are 15.7 ± 0.197 mm long and 19.5 ± 0.280 mm wide (mean ± SE) but, while pistillate, they are 14.6 ± 0.365 mm long and 17.8 ± 0.318 mm wide. Flowers of both sexes remain open for 8 h [55]. Female flowers emerge first at the basal part of the inflorescence, while male flowers emerge later in the apical section once the female flowers die, so it is rare to find inflorescences with simultaneously functional male and female flowers [55] (Figure 5B). Nectar is produced in the base of the gynoecium (females) and on the insertion of the stamens (males). The surface of the exposed seeds has a rough appearance (Figure 5C). The fruits are green when ripe and globose with a 0.8–1.5 cm diameter, containing up to 1500 seeds, which are dispersed by gravity or wind [54,56].

### 4.3. Characterization of Florivory

To record florivorous insects on the flowers of *S. lancifolia*, we walked extensively throughout the wetland. The behavior of the insects found in buds and both female and male flowers was observed. Individuals of the insects recorded that consume the petals of the female and male flowers were collected for determination. Voucher specimens were deposited in the IEXA entomological collection of the Instituto of Ecología, A.C.

We quantified the natural florivory in male and female flowers of *Sagittaria Lancifolia* and assigned damage categories according to the area of tissue removed from the petals, following the foliar herbivory methodology of Dirzo and Domínguez [57]. The damage categories employed were: (0) flowers intact, (1) flowers with 1–6% damage, (2) flowers with 7–12% damage, (3) flowers with 13–25% damage, (4) flowers with 26–50% damage, and (5) flowers with 51–100% damage (Figure 6). A similar methodology has been used to characterize florivory in other studies [6].

The amount of florivory damage (Florivory Index, FI) present in the male and female flowers was calculated using the same equation from the herbivory index of Dirzo and Domínguez [57] as follows: (1)$$FI=(X_{i}\times n_{i})/N,$$
where *X_i_* is the florivory category (damage score, 0 to 5), *n_i_* is the number of flowers in the category *X_i_* evaluated, and *N* is the total number of flowers. The FI was then expressed as a percentage of florivory following the methodology proposed by Ruiz-Guerra et al. [58], which is based on a cubic linear model with intercept set to zero:(2)$$Florivory\%=5.6131\times FI-2.4505\times {FI}^{2}+0.8691\times {FI}^{3}N,$$

The premise for this model is that the expected florivory (EF) for each *X_i_* score of damage is given by:(3)$$EF(\%)=\frac{F_{i\;max}-(F_{i\;max}\;–\;F_{i\;min})}{2},$$
where *F_i max_* and *F_i min_* correspond to the upper and asymptotic lower limits of florivory, respectively, of each of the *X_i_* scores of damage. Based on this equation, the expected percentages of florivory are 0, 3, 6, 18.5, 37.5, and 75% for the scores 0 to 5, respectively. A Wilcoxon test was performed to detect differences in the percentage of flower damage between male and female flowers. Data were arcsine transformed and normality explored before analysis. Analyses were conducted using PAST 4.11 [59].

### 4.4. Floral Visitors on Damaged Flowers

To assess the effect of the different levels of florivory on the frequency of floral visitors, we selected inflorescences with at least 6 flowers of the same sexual expression (female or male) of different damage categories and one undamaged. When all damage categories were not present on the flowers of one inflorescence, we made artificial damage with scissors to obtain the missing categories of florivory. On the other hand, when one inflorescence did not have enough flowers to complete the six categories of damage, the nearest plant (not more than 20–25 cm apart) was used to complete the experimental set of flowers, following the procedures of selection and artificial damage described above. The remaining flowers on the inflorescence were cut, so only the experimental set of flowers was available to floral visitors. This allowed us to avoid any variation due to differences in the size of inflorescence or floral display. The groups of flowers of each damage category were marked with cotton threads of a different color, which allowed quick identification of the level of damage when the floral visitors arrived.

Two observers recorded the frequency of floral visitors for 6 days during the period from October to November 2011. Daily visitor records were taken from 08:00 to 15:00 h, with periods of 15 min of observation and 15 min of rest. The flowers of each sex were observed for a total of 48 h. We considered a true visit to have occurred when the insects made physical contact with the reproductive structures (stigma or stamens). Specimens of each species of floral visitor were caught using an entomological net and preserved individually; beetles, flies and bees were placed in alcohol (70%), and butterflies in wax bags, with subsequent mounting with entomological pins. Identification was performed to the lowest taxonomic level possible, although, for the analysis, insects were grouped by order. Voucher specimens were deposited in the entomological collection of Instituto de Ecología, A.C. (IEXA). Differences in flower visitor frequencies among damage classes and flower sexual expressions were explored with a two-way ANOVA, in which flower sex and florivory damage category were fixed factors. Tukey’s post hoc multiple comparison tests allowed us to identify differences between florivory damage categories and flower sexes. Analyses were conducted using PAST 4.11 [59].

### 4.5. Effect of Florivory on Female Reproductive Success

Newly opened female flowers with natural or artificial damage were selected and labeled to differentiate the damage categories as described above (29–47 flowers per category of florivory damage, a total of 211 flowers). Flowers were exposed to open pollination and inspected frequently to remove herbivores to avoid further damage. Reproductive success (fruit set) was estimated for each damage category by the ratio between the number of fruits produced and the initial number of experimental flowers. The ripe fruits were collected individually in tagged paper bags and weighed (Ohaus Scout Pro 400 g, accuracy of 0.01 g). We estimated the number of seeds per fruit using the following procedure: five fruits were randomly selected per damage category, each fruit was weighed, and its seeds were extracted manually, counted, and weighed before storing them individually in labeled containers. Three samples of 100 seeds of each fruit were then weighed on an analytical balance (Sartorius BP221D^®^, accuracy 1 mg), and the average weight of 100 seeds per florivory damage category was obtained. The seed number of each fruit of each *i*-damage category was obtained by multiplying 100 (standardized seed number weighed) by the weight of total seeds of the fruit from the *i*-category, and divided by the average weight of the 100 seeds from the same damage category. After the Shapiro–Wilk test for data normality, we used a Kruskal–Wallis non-parametric test to find out if there were differences in the fruit weight and number of seeds among the categories of florivory damage. Mann–Whitney pairwise tests then allowed us to identify the differences between categories. Analyses were performed using PAST 4.11 software [59].

### 4.6. Evaluation of the Effect of Flower Manipulation on Fruit and Seed Set

To determine whether or not the fruit set recorded in the category with the highest floral damage (51–100%) was a consequence of the flower manipulation when the plant was artificially damaged [48], 16 inflorescences of different plants with newly opened female flowers were randomly selected. Two flowers per inflorescence, one undamaged and one with the highest artificial damage, were hand-pollinated with a mixed pollen load from three different male flowers of different inflorescences. Experimental flowers were monitored frequently throughout the daytime to prevent florivory. All flowers of both treatments were tagged, and fruits were harvested when ripe. Fruit weight and number of seeds were obtained as described above. After the normality test (Shapiro–Wilk), we used a *t*-test analysis to determine whether there were significant differences in the weight and number of seeds between treatments. The analyses were performed using PAST 4.11 software [59].

## 5. Conclusions

The consequences of florivory for pollination and reproductive success are variable and depend largely on specialization or generalization in the plant–pollinator interaction and the intensity of damage caused by the florivores. To our knowledge, this is the first study to evaluate the direct and indirect effects of florivory in a monoecious species under natural conditions. Our results clearly show the negative effect of the intensity of florivory by *Tanisphirus lemnae* (Curculionidae) on the different components of the reproductive success of *Sagittaria lancifolia*, although further research is required to fully understand the processes involved. Further studies are also needed to understand the effects of different levels of florivory on the germination response of seeds produced by female flowers (maternal component) and on the pollen load exportation of male flowers (paternal component), which currently remain unknown.

## Figures and Tables

**Figure 1 plants-13-00547-f001:**
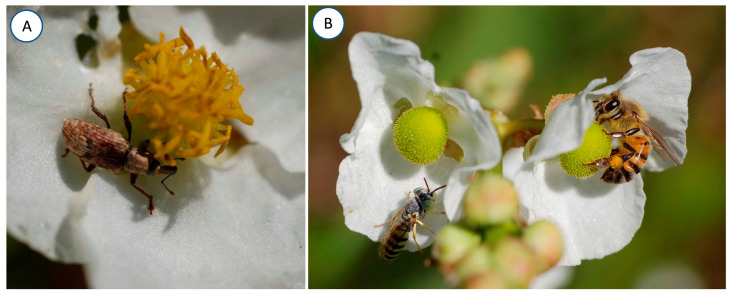
(**A**) Male flower of *Sagittaria lancifolia* with the florivorous weevil *Tanysphyrus lemnae*. (**B**) Female flowers of *Sagittaria lancifolia* visited by *Apis mellifera* and one wasp (Crabronidae).

**Figure 2 plants-13-00547-f002:**
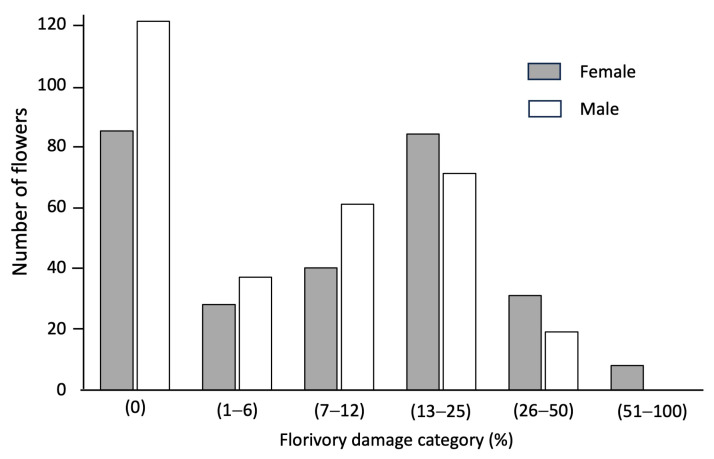
Numbers of female and male flowers of *Sagittaria lancifolia* in each florivory damage category recorded in a wetland of La Mancha, Veracruz, Mexico.

**Figure 3 plants-13-00547-f003:**
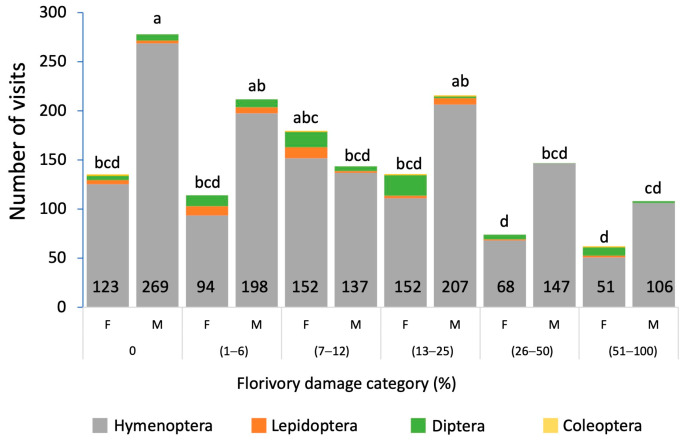
Number of visits and taxonomic orders of the floral visitors to female (F) and male (M) flowers of *Sagittaria lancifolia* across the different florivory damage categories recorded in a wetland at La Mancha, Veracruz, Mexico. Different letters in the bars indicate significant differences.

**Figure 4 plants-13-00547-f004:**
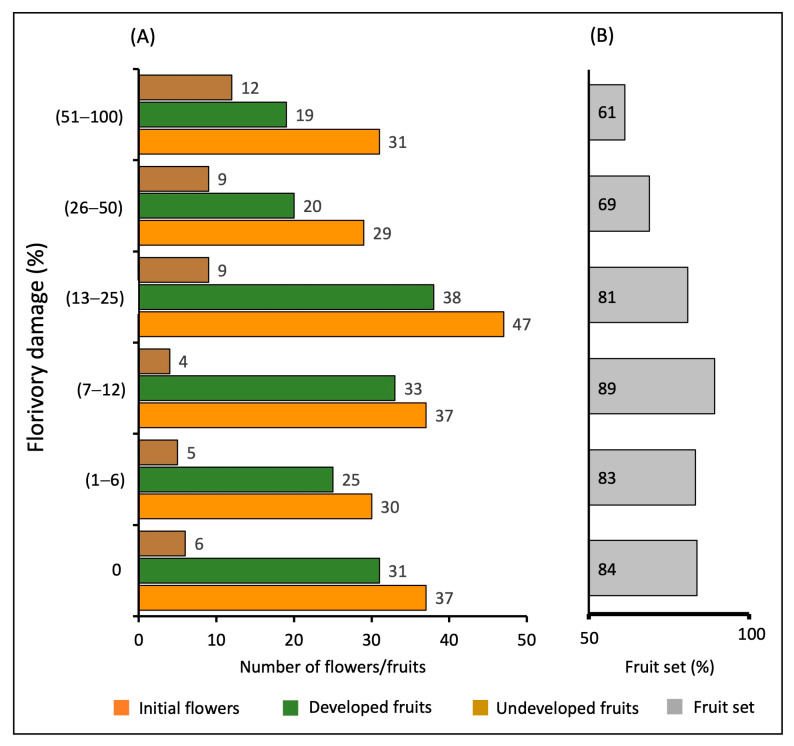
(**A**) Number of initial flowers (orange), developed fruits (green), and undeveloped fruits (brown), and (**B**) fruit set (gray) recorded in *Sagittaria lancifolia* in different florivory damage categories.

**Figure 5 plants-13-00547-f005:**
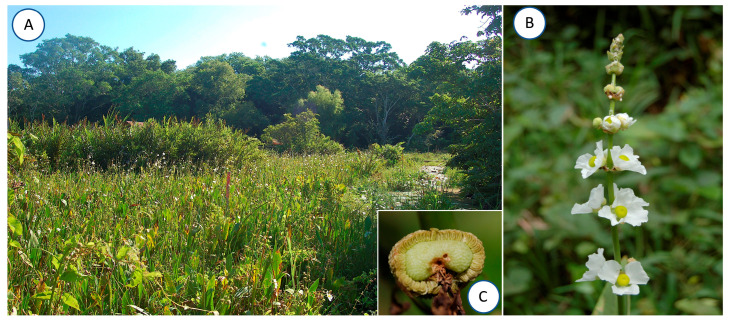
(**A**) View of the wetland studied at La Mancha, Veracruz, Mexico. (**B**) Inflorescence of *Sagittaria lancifolia* showing female flowers. (**C**) Fruit with seeds visible.

**Figure 6 plants-13-00547-f006:**
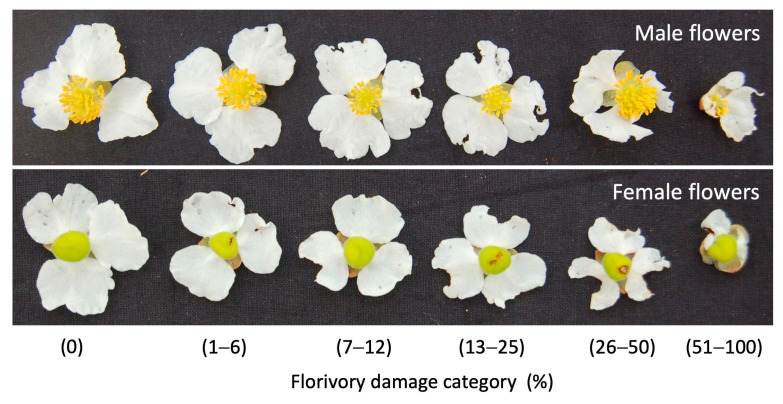
Categories of damage identified in male and female flowers in *S. lancifolia*.

**Table 1 plants-13-00547-t001:** Weight of the fruits and seeds, and the number of seeds per fruit developed in the female flowers of *Sagittaria lancifolia* in different florivory damage categories exposed to natural pollination in a wetland at La Mancha, Veracruz, Mexico. Different letters within columns indicate significant differences (Kruskal–Wallis *p* < 0.01, and Mann–Whitney pairwise test *p* < 0.01). For all variables, mean values ± SE are presented.

Florivory Damage Category (%)	Fruit Weight (g)	Seed Weight (g)	Number of Seeds per Fruit
0	0.237 ± 0.019 ^a^	0.213 ± 0.018 ^a^	1287.3 ± 107.7 ^a^
1–6	0.237 ± 0.023 ^a^	0.214 ± 0.021 ^a^	1080.3 ± 106.3 ^ac^
7–12	0.188 ± 0.012 ^b^	0.169 ± 0.011 ^b^	910.7 ± 60.0 ^bcd^
13–25	0.120 ± 0.010 ^b^	0.107 ± 0.010 ^b^	859.0 ± 80.8 ^bc^
51–100	0.146 ± 0.014 ^b^	0.132 ± 0.013 ^b^	791.9 ± 77.9 ^bc^

**Table 2 plants-13-00547-t002:** Weight and number of seeds per fruit developed in the female flowers of *Sagittaria lancifolia* obtained from manually pollinated flowers in a wetland at La Mancha, Veracruz, Mexico. For all variables, mean values ± SE are presented.

Flower Treatment	Seed Weight (g)	Seed Number per Fruit
Undamaged	0.208 ± 0.027	1251.7 ± 143.2
Florivory	0.205 ± 0.031	1164.1 ± 166.9

## Data Availability

All relevant data are within the manuscript.

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
