# Peer review of "Effects of Florivory on Floral Visitors and Reproductive Success of Sagittaria lancifolia (Alismataceae) in a Mexican Wetland"

_plants, 2024, doi:10.3390/plants13040547_

Round 1

Reviewer 1 Report

Comments and Suggestions for Authors

The study analyzes the effect of florivory on Sagittaria lancifolia reproductive success and pollinators visitors. Is a laborious and thorough study with interesting results; the effects are significant for the plant-pollinator interaction and the Sagittaria's flowers morphology.

I have a suggestion/question  for the authors: based on these results maybe you'll consider any positive effect of florivory considering the population level of Sagittaria? The florivory is a category of herbivory so are to be expected "hidden" effects both on the "prey" population and consumer population. Maybe a further study will achieve this topic. 

Author Response

Comment: The study analyzes the effect of florivory on Sagittaria lancifolia reproductive success and pollinators visitors. Is a laborious and thorough study with interesting results; the effects are significant for the plant-pollinator interaction and the Sagittaria's flowers morphology.

Comment: I have a suggestion/question for the authors: based on these results maybe you'll consider any positive effect of florivory considering the population level of Sagittaria? The florivory is a category of herbivory so are to be expected "hidden" effects both on the "prey" population and consumer population. Maybe a further study will achieve this topic.

Response: Great question. Our results show negative effects directly related to increased levels of florivory. We suggest in the Conclusions section that further studies about seed germination can help us better understand all the effects of this interaction. Despite the reduction in weight and number of seeds, the germination rate will increase as compensation response to florivory. Understanding the effect of florivory on germination could help understand aspects of population dynamics, such as recruitment. Likewise, it would be necessary to design experiments in sites where plants receive different intensities of florivory and thus evaluate different population growth parameters.

On the other hand, we experimentally demonstrated that the number of seeds and seed weight between damaged and undamaged flowers hand-pollinated, were not different (Table 2). This suggests that pollen load could be another important factor in the differences found in our study. 

We address this idea a bit in the conclusions section.

Reviewer 2 Report

Comments and Suggestions for Authors

This is a well-designed and implemented study that enhances understanding of the influence of weevil florivory upon seed size and production in Sagittaria lancifolia.  The English is excellent, it is well-written, and data are presented in a clearly understandable format.  The manuscript is acceptable to publish in its current form.

Since seed weight was negatively influenced by florivory, is there insight as to how this might influence seed viability?  Has there been any research upon viability in seeds of different sizes and weights?

In the Materials and Methods section lines 237, there are typographical errors resulting in the misspelling of the genus name and Thyphaceae instead of Typhaceae.

Since Saggitaria lancifolia reproduces clonally, does florivory influence that means of population expansion – by increasing clonal activity, for example?  Has there been observation of lowering seed production or florivory resulting in larger leaves or increased reliance upon greater clonal activity?

Author Response

Comment: This is a well-designed and implemented study that enhances understanding of the influence of weevil florivory upon seed size and production in Sagittaria lancifolia. The English is excellent, it is well-written, and data are presented in a clearly understandable format. The manuscript is acceptable to publish in its current form.

Response: We appreciate your kind comment about our study

Comment: Since seed weight was negatively influenced by florivory, is there insight as to how this might influence seed viability? Has there been any research upon viability in seeds of different sizes and weights?

Response: Thank you they are good questions. There are many studies evaluating seed germination in Sagittaria spp. but no one related to florivory. These studies explore the effects of several environmental conditions like temperature, water level, salinity, oxygen, and soil characteristics among others (v.gr., Keddy & Constable 1986, Delsalle & Blum 1994, Dai et al 2018, Kenow et al. 2018; Ozaki et al 2018). However, as far as we know there are not any studies evaluating how florivory can influence seed viability and germination. We developed a few sentences at the end of the Conclusion section (L369-372) about that. We think knowing the seed responses to florivory lets us understand more about the population dynamics of this species.

References

Delesalle VA, Blum, S. Variation in germination and survival among families of Sagittaria latifolia in response to salinity and temperature. Int J Plant Sci 155(2). https://doi.org/10.1086/297158

Keddy PA, Constabel P. 1986. Germination of Ten Shoreline Plants in Relation to Seed Size, Soil Particle Size and Water. J Ecol 74, 133-141

Kenow KP, Gray BR, Lyon JE. 2018. Flooding tolerance of Sagittaria latifolia and Sagittaria rigida under controlled laboratory conditions. River Res Applic 34,1024–1031. https://doi.org/10.1002/rra.3337

Ozaki Y, Shimono Y, Tominaga T. 2018. Germination characteristics of Sagittaria trifolia. Weed Biol Manag 18, 160-166. https://doi.org/10.1111/wbm.12162

Comment: In the Materials and Methods section lines 237, there are typographical errors resulting in the misspelling of the genus name and Thyphaceae instead of Typhaceae.

Response: Thank you for noting this error. The correction was made. The change is in the same line in the new file version loaded.
